# *Sporothrix brasiliensis* Causing Atypical Sporotrichosis in Brazil: A Systematic Review

**DOI:** 10.3390/jof10040287

**Published:** 2024-04-13

**Authors:** Vanice Rodrigues Poester, Melissa Orzechowski Xavier, Lívia Silveira Munhoz, Rossana Patricia Basso, Rosely Maria Zancopé-Oliveira, Dayvison Francis Saraiva Freitas, Alessandro Comarú Pasqualotto

**Affiliations:** 1Programa de Pós-Graduação em Ciências da Saúde, Faculdade de Medicina (FAMED), Universidade Federal do Rio Grande (FURG), Rio Grande 96200-190, Rio Grande do Sul (RS), Brazil; vanicerp@gmail.com (V.R.P.); melissaxavierfurg@gmail.com (M.O.X.); liviasmunhoz@gmail.com (L.S.M.); rossanabasso3@gmail.com (R.P.B.); 2Mycology Laboratory of FAMED-FURG, Rio Grande 96200-190, RS, Brazil; 3Hospital Universitário Dr. Miguel Riet Correa Jr., FURG/Empresa Brasileira de Serviços Hospitalares (EBSERH), Rio Grande 96200-190, RS, Brazil; 4Instituto Nacional de Infectologia Evandro Chagas, Fundação Oswaldo Cruz, Rio de Janeiro 21040-360, Rio de Janeiro (RJ), Brazil; rosely.zancope@ini.fiocruz.br (R.M.Z.-O.); dayvison.freitas@ini.fiocruz.br (D.F.S.F.); 5Molecular Biology Laboratory, Hospital Dom Vicente Scherer, Santa Casa de Misericórdia de Porto Alegre, Porto Alegre 90020-090, Brazil; 6Medicine Department, Universidade Federal de Ciências da Saúde de Porto Alegre, Porto Alegre 90035-075, RS, Brazil

**Keywords:** zoonosis, ocular, nasal, hypersensitivity, osteoarthritis, pulmonary, meningeal disease

## Abstract

Zoonotic sporotrichosis, a subcutaneous mycosis caused by *Sporothrix brasiliensis*, has become hyperendemic and a serious public health issue in Brazil and an emerging disease throughout the world. Typical sporotrichosis is defined as fixed or lymphocutaneous lesion development, however, reports of atypical presentations have been described in hyperendemic areas, which may result in a worse prognosis. Thus, considering an increase in atypical cases and in more severe extracutaneous cases and hospitalizations reported in Brazil, we aimed to perform a systematic review to search for hypersensitivity reactions (HRs) and extracutaneous presentations associated with zoonotic sporotrichosis. A systematic review was performed, following the PRISMA guidelines to search for atypical/extracutaneous cases (mucosal, osteoarthritis, HRs, pulmonary, meningeal) of zoonotic sporotrichosis. A total of 791 published cases over 26 years (1998–2023) in eleven Brazilian states were reviewed. Most cases corresponded to a HR (47%; n = 370), followed by mucosal (32%; n = 256), multifocal (8%; n = 60), osteoarthritis (7%; n = 59), meningeal (4%; n = 32), and pulmonary (2%; n = 14) infections. When available (n = 607), the outcome was death in 7% (n = 43) of cases. Here, we show a frequent and worrisome scenario of zoonotic sporotrichosis in Brazil, with a high and dispersed incidence of atypical/extracutaneous cases throughout the Brazilian territory. Therefore, educational measures are necessary to make health professionals and the overall population aware of this fungal pathogen in Brazil as well as in other countries in the Americas.

## 1. Introduction

*Sporothrix brasiliensis* is one of the major causes of sporotrichosis, a neglected implantation mycosis that has become a significant public health problem in South America, especially in Brazil. Thousands of cases of zoonotic sporotrichosis have been described, and, in the last decade, this disease has spread all over the Brazilian territory [1,2,3]. Not restricted to Brazil anymore, nowadays *S. brasiliensis* is also an emergent pathogen in other South American countries, with autochthonous cases being described in Chile, Argentina, Paraguay, and Uruguay. Furthermore, cases of zoonotic sporotrichosis have been reported in recent years in other continents (North America and Europe) where patients were infected by cats imported from Brazil [4,5,6,7,8,9]. 

Domestic cats are the main victims of sporotrichosis caused by *S. brasiliensis* and the main source of the dissemination of this fungal species. Affected animals usually develop a severe manifestation of the disease, often with disseminated cutaneous and nasal lesions, which may potentially lead to death [10,11]. The high fungal burden found in feline cutaneous lesions explains how easily *S. brasiliensis* can be transmitted by bites or scratches to other cats or humans [12]. Since feline sporotrichosis outbreaks are increasing in number and spreading geographically, zoonotic sporotrichosis nowadays involves several epidemiological scenarios, including (i) areas of hyperendemicity (in many regions of Brazil and some Argentinian provinces); (ii) large outbreaks (in distinct regions of South America); and (iii) occasional cases occurring in other continents, resulting in an increased awareness.

Although most zoonotic sporotrichosis is clinically evidenced by the typical known fixed or lymphocutaneous lesions, reports of atypical presentations have been described in hyperendemic areas, which may result in a worse prognosis [13,14,15,16,17,18,19,20,21]. Mucosal involvement, hypersensitivity reactions (HRs), osteoarthritis, and pulmonary and meningeal involvement are highlighted as atypical presentations of sporotrichosis [17,18,19,20,21,22]. *S. brasiliensis* is more likely to cause atypical clinical manifestations of sporotrichosis than the sapronotic species, *S. schenckii* and *S. globosa* [14].

HRs in zoonotic sporotrichosis can occur as (i) Sweet’s syndrome, characterized as painful erythematous papules, plaques, or nodules with neutrophil infiltrate, associated with fever; (ii) erythema nodosum, defined as the appearance of painful erythematous subcutaneous nodules in the lower limbs, with fever, arthralgia, and myalgia; (iii) erythema multiforme, an acute inflammatory reaction, showing erythematous plaques on the skin or mucosa, typically target-shaped; and (iv) arthritis, characterized as an aseptic inflammatory articular process resulting in pain and edema, frequently affecting the knees, wrists, elbows, or ankles [17,23,24,25,26,27,28]. Although atypical, the HRs fortunately have a good prognosis, but they require a correct diagnosis by healthcare professionals [17,23].

In mucosal involvement, ocular sporotrichosis is described in a diversity of ophthalmologic patterns, which occur as ocular adnexal or intraocular infections. Sporotrichosis in ocular adnexa is subclassified as eyelid involvement, granulomatous conjunctivitis, Parinaud syndrome, or disorders of the lacrimal system (dacryocystitis). Intraocular sporotrichosis can be presented as endophthalmitis, granulomatous uveitis, scleritis, retinitis, choroiditis, or iridocyclitis [15,29,30,31,32]. Nasal involvement is another manifestation of mucosal zoonotic sporotrichosis, which varies from mild to severe, progressing to a perforation of the septum [21].

Sporotrichosis in bones or causing septic articular involvement occurs mostly as a consequence of the extension of a cutaneous lesion, through the classical traumatic route, or due to *Sporothrix* sp. hematogenous dissemination [33]. Similarly, pulmonary and meningeal infection can follow *Sporothrix* conidia inhalation, after fungal dissemination [18,19,34,35,36]. Severe atypical presentations of sporotrichosis are often associated with comorbidities, such as AIDS, alcoholism, chronic obstructive pulmonary disease, or diabetes [13,18,37].

Considering the increase in the reports of these atypical manifestations in patients with zoonotic sporotrichosis, which can lead to a higher rate of hospitalization and a worse prognosis [3,9,23,38], we compiled data from all cases reported and/or series of cases published in the literature in a unique and combined approach, to better understand their impact. Therefore, here we perform a systematic review of the literature on zoonotic sporotrichosis involving HR and extracutaneous presentations.

## 2. Materials and Methods

A systematic review was performed, following the PRISMA guidelines [39], using the databases Pubmed, SciELO, Web of Science, and LILACS, which were last consulted in February 2024. Aiming to expand the search, references of the selected articles were reviewed to find additional articles. Descriptors were (extracutaneous and sporotrichosis) OR (epidemiological and *Sporothrix* and *brasiliensis*) OR (disseminated and *Sporothrix* and *brasiliensis*) OR (pulmonary and *Sporothrix* and *brasiliensis*) OR (ocular and *Sporothrix* and *brasiliensis*) OR (nasal and *Sporothrix* and *brasiliensis*) OR (hypersensitivity and *Sporothrix* and *brasiliensis*) OR (osteoarthritis and *Sporothrix* and *brasiliensis*) OR (osteomyelitis and *Sporothrix* and *brasiliensis*) OR (meningeal and *Sporothrix* and *brasiliensis*). No automation tools were used to search articles, and two researchers performed the search independently, reducing the bias.

We included cases of extracutaneous sporotrichosis (mucosal, osteoarthritis, pulmonary, meningeal) and HRs. Both cases of proven (with *Sporothrix* spp. isolation in culture from the primary site of infection) and probable (patients with clinical–epidemiological characteristics of zoonotic sporotrichosis) sporotrichosis were included [40]. Each case was categorized as (i) mucosal: patients with ocular (adnexal and/or intraocular), nasal and/or oral involvement; (ii) osteoarthritis: patients with osteomyelitis and/or septic arthritis; (iii) a HR: patients showing Sweet’s syndrome, erythema nodosum, erythema multiforme, or aseptic arthritis in consequence of *Sporothrix* spp. infection; (iv) pulmonary: patients with lung involvement; (v) or meningeal: patients with central nervous system involvement. Cases of sporotrichosis were also classified as unifocal (involving a single site or two contiguous sites of infection/type of atypical presentation) or multifocal (two or more non-contiguous sites). 

Articles were initially selected by title, then by abstract, without language restriction. Eligible articles were read in full and included when they met the inclusion criteria. Criteria for inclusion were articles (i) from 1990 (beginning of zoonotic outbreaks) to December 2023 and (ii) describing atypical sporotrichosis manifestations (mucosal, osteoarthritis, HR, pulmonary, or meningeal). Review articles and case series with no data regarding the site of infection were excluded.

Cases were grouped and analyzed in each category (HRs, mucosal, osteoarthritis, pulmonary, meningeal) regarding sex, age, dispersion in time and location, primary site of infection and sites of dissemination, presence of comorbidities, and outcome. Data compilation and analyses were performed with the Excel software (2013 version, Microsoft Corporation^®^, Redmond, WA, USA). Geographical analyses were performed using Google Maps (https://maps.google.com/, accessed on 28 February 2024) (Google^®^, Mountain View, CA, USA) superimposed on a world map, generating a Geographic Information System (GIS). QGIS software version 3.30.0 (Open Source Geospatial Foundation—OSGeo, Anchorage, AK, USA) was used to analyze the geographical distribution of cases.

## 3. Results

### 3.1. Article Searching

A total of 442 articles were selected for analysis (404 from databases and 38 from references of articles), and 206 were excluded due to duplicate records, with no other reason for excluding articles. After reviewing titles and abstracts, no additional records were retrieved, and 89 articles were selected for full-text reading. Of these, 64 articles were included in our systematic review (Figure 1). 

The first case of atypical zoonotic sporotrichosis was published in 2002. Nearly half of the articles (55%; 35/64) were published in the last five years (2019–2023), which correspond to an increase of 700% in comparison with the number of publications from the first 5 years of our review (Figure 2) [13,14,15,17,18,19,20,21,22,24,25,26,27,28,29,31,32,34,35,36,37,41,42,43,44,45,46,47,48,49,50,51,52,53,54,55,56,57,58,59,60,61,62,63,64,65,66,67,68,69,70,71,72,73,74,75,76,77,78,79,80,81,82,83].

### 3.2. Extracutaneous and HR Cases

A total of 791 cases of patients showing atypical zoonotic sporotrichosis were described in the 64 articles included in this review, all of them in patients from Brazil. Geographic distribution showed the occurrence throughout eleven Brazilian states, with 88% (n = 695) of cases occurring in Rio de Janeiro. Other states with the description of patients with atypical zoonotic sporotrichosis were Minas Gerais (5%; n = 36), Paraná (3%; n = 23), Pernambuco (1%; n = 11), São Paulo (1%; n = 8), Rio Grande do Sul (<1%; n = 7), Paraíba (<1%; n = 7), Espírito Santo, Bahia, Distrito Federal, and Rio Grande do Norte (<1%; n = 4; one case from each of the last four states described) (Figure 3).

Most patients, 66% (453/689), were female, and the mean age was 38 years old (ranging from 2 to 80, standard deviation: 18; n = 682). More than 90% of cases involved close contact with cats, and in 98% a proven diagnosis of sporotrichosis was achieved by the isolation of the fungus in mycological culture. 

Into the group of patients with unifocal atypical zoonotic sporotrichosis (n = 731), a HR was the main presentation occurring in 51% of the patients (n = 370), with mucosal involvement being the second mainly atypical condition, occurring in 35% (n = 256) of the patients. Patients with more severe patterns of atypical zoonotic sporotrichosis totalized 105, including 8% (n = 59) with osteoarthritis, 4% (n = 32) with meningeal involvement, and 2% (n = 14) with pulmonary involvement. The other 60 patients had multifocal disease, often (53 patients) involving two sites/types of presentation: osteoarthritis + mucosal (n = 32); osteoarthritis + pulmonary (n = 14); osteoarthritis + meningeal (n = 3); mucosal + meningeal (n = 2); and meningeal + pulmonary (n = 2). There were six patients with three sites: osteoarthritis + meningeal + pulmonary (n = 3); osteoarthritis + meningeal + mucosal (n = 2); and osteoarthritis + mucosal + pulmonary (n = 1). One patient showed osteoarthritis + mucosal + meningeal + pulmonary involvement (Table 1 and Table 2).

Comorbidities were described in 53% (110/209) of the total of patients for whom these data were available. The predominant condition was HIV infection (n = 74), followed by systemic arterial hypertension (n = 20), alcohol abuse (n = 19), and diabetes (n = 12). Other comorbidities described were tuberculosis (n = 7), renal transplantation or chronic renal failure (n = 4), chronic obstructive pulmonary disease (n = 3), asthma (n = 2), bacteremia (n = 2), tobacco abuse (n = 2), corticosteroid use (n = 2), hepatitis C infection (n = 2), malnutrition (n = 2), cytomegalovirus infection (n = 2), and epilepsy (n = 2). Additional conditions included (n = 1 each) drug abuse, chronic anemia, bronchiectasis, post-COVID-19 with asthma, deep vein thrombosis, pneumocystosis, neurotoxoplasmosis, nocardiosis, sarcoidosis, dyslipidemia, Takayasu arthritis, lepra, fibromyalgia, hypothyroidism, hepatic steatosis, benign prostatic hyperplasia, and/or pulmonary arterial hypertension. Most (62%; 18/29) patients with a HR had no underlying condition, while in the mucosal form of sporotrichosis 45% (29/65) of the patients had comorbidities. In the more severe forms of sporotrichosis (osteoarthritis, meningeal, pulmonary, and multifocal), 94% (108/115) of patients had comorbidities.

#### 3.2.1. HR Cases

Erythema nodosum or multiforme represented 88% of the HRs (327/370), followed by Sweet’s syndrome in 10% (35/370) and arthritis in 2% (8/370) (Table 1). The patients being cured was the outcome in all zoonotic sporotrichosis cases that developed HRs (n = 318) [14,17,24,25,26,27,28,42,47,54,69,76,81]. 

#### 3.2.2. Mucosal Cases

Mucosal cases were predominantly ocular (86%; 221/256), followed by nasal (11%; 28/256) and oral (<1%; 2/256). Four patients (2%) had oral and nasal sporotrichosis associated, and one patient (<1%) showed ocular and nasal lesions. Data of the pattern of ocular disease were available from 199/221 patients, 158 (83%) with only one pattern (conjunctivitis, n = 123; eyelids, n = 27; dacryocystitis, n = 7; retinal granuloma, n = 1), 39 presenting two associated patterns (conjunctivitis + Parinaud syndrome, n = 23; conjunctivitis + dacryocystitis, n = 5; conjunctivitis + eyelids, n = 3; Parinaud syndrome + eyelids, n = 3; Parinaud syndrome + dacryocystitis, n = 3; or dacryocystitis + eyelids, n = 2), and two patients showing three associated patterns (conjunctivitis + eyelids + dacryocystitis, n = 2). One patient presented ocular involvement (Parinaud syndrome + dacryocystitis) together with a HR.

Mucosal involvement occurred as a primary disease in the majority of the patients (92%, n = 230/251), while the remaining (8%, n = 21) developed mucosal sporotrichosis due to a hematogenous dissemination of *Sporothrix* spp. Even though all such patients were cured, 12% (23/193) had sequelae of the mycosis, such as chronic dacryocystitis, corneal changes, cutaneous fistula, lagophthalmos, ectropion, entropion, pannus 180°, symblepharon, conjunctival fibrosis, paracentral leucoma or eyelid retraction (ocular), and hyperrhynolalia and retraction of the right ala nasi (nasal) (Table 1) [15,20,21,29,31,32,41,42,47,48,52,53,57,60,61,63,64,65,67,70,71,73,74,75,78,79,81,82]. 

#### 3.2.3. Osteoarthritis Cases

Osteoarthritis as an extracutaneous manifestation of zoonotic sporotrichosis was reported in 59 patients, 42 as osteomyelitis and 17 as septic arthritis. Information on the site of osteoarthritis was available for 46 patients. This most frequently involved the hands (56%; n = 26), followed by upper and/or lower limbs (20%; n = 9), feet and/or ankles (13%; n = 6), knees (7%; n = 3), elbows (2%; n = 1), and clavicles (2%; n = 1). Fifty-five percent (27/49) occurred after fungal hematogenous dissemination, with the bone involvement a consequence (contiguity infection) of a cutaneous lesion in the other 22 patients (45%); in 10 patients these data were unavailable. The outcomes were described in case reports of 26 patients, with 85% (22/26) being cured and 15% (4/26) dying. From the 22 cured patients, sequelae (total or partial amputation) occurred in 14% (3/22) (Table 1) [22,37,46,50,56,76,79,80,81,83]. 

#### 3.2.4. Pulmonary Cases

Pulmonary zoonotic sporotrichosis occurred as a primary disease in 71% of patients (10/14), and in 29% (4/14) it occurred because of a hematogenous fungal dissemination. The infection resulted in cavitary nodules (n = 4), bronchiectasis (n = 2), lung infiltrate (n = 1), pleural effusion (n = 1), and/or fibrosis (n = 1) (data available for eight patients only). In 93% (13/14) of cases, *Sporothrix* spp. was isolated from respiratory samples in mycological cultures (Table 1). In total, 60% of these patients were cured (3/5), including two patients with disseminated disease and one patient with primary lung involvement [18,37,49,51,59,66,68,72,77,81].

#### 3.2.5. Meningeal Cases

Meningeal involvement was reported in 32 cases, the majority (91%, 29/32) occurring in immunosuppressed patients, with sporotrichosis disseminating to the central nervous system. Underlying conditions for these patients included HIV infection (n = 25), drug/alcohol abuse (n = 3), liver transplantation (n = 1), leprosy (n = 1), or chronic steroid use (n = 1), with an overlap of diseases in two cases (drug/alcohol abuse and liver transplantation; leprosy and chronic steroid use). The three (9%) other patients with neurological sporotrichosis as a primary disease were immunocompetent, and the diagnosis was confirmed by the isolation of *Sporothrix* spp. from the cerebrospinal fluid. Death was the predominant outcome (70%, 21/30), with one immunocompetent individual, and 30% (9/30) of patients were cured, including two immunocompetent patients, with sequelae (ataxia and extrinsic ocular motor paresis) reported in one (11%; 1/9) [19,34,35,36,43,44,45,47,55,62].

#### 3.2.6. Multifocal Cases

Among 60 patients with multifocal atypical sporotrichosis, 88% (53/60) presented two sites of involvement, while others had three (10%; 6/60) or four (2%; 1/60) non-contiguous sites of infection (Table 1). The predominant outcome of multifocal cases was the patient being cured (54%; 19/35); sequelae occurred at a rate of 11% (2/19), associated with bone involvement; and death occurred in 46% (16/35) of patients. It was reported that 68% and 67% of patients with osteoarthritis plus mucosal or pulmonary involvement were cured, respectively. However, all patients with the following presentations died: osteoarthritis + meningeal; meningeal + pulmonary; osteoarthritis + meningeal + pulmonary; osteoarthritis + meningeal + mucosal; and osteoarthritis + mucosal + pulmonary [18,19,21,22,37,47,58,71,79] (Table 2).

## 4. Discussion

In this systematic review, we summarized ~800 atypical zoonotic sporotrichosis cases described in the literature, with an exponential increase in the number of papers reporting these atypical cases of zoonotic sporotrichosis in the last five years of study, which coincides with the significant expansion of the disease throughout the Brazilian territory [84]. 

All the atypical cases of zoonotic sporotrichosis occurred in Brazil, which is considered the epicenter of the disease in the world [9], and the majority occurred in the state of Rio de Janeiro, where this mycosis has a hyperendemic status, in a scenario that began as an outbreak at the end of the 1990s [3,42]. Besides Rio de Janeiro, atypical zoonotic sporotrichosis cases were also reported in patients from ten additional Brazilian states. Two of these states (Rio Grande do Sul and São Paulo) are nowadays hyperendemic areas for sporotrichosis, starting with outbreak reports since the 2000s [76,85,86,87], and in eight other states (Paraná, Distrito Federal, Minas Gerais, Pernambuco, Bahia, Rio Grande do Norte, Paraíba, and Espírito Santo) outbreaks of *S. brasiliensis* infection have been reported in the last five/ten years [79,88,89,90,91,92,93].

Thus, we can speculate that, in general, atypical cases of zoonotic sporotrichosis are not directly related to areas with a high endemicity already established for long periods. Instead, atypical cases can occur from the beginning of *S. brasiliensis* reports, which can be attributed to fungi strain characteristics, considering that they exhibit distinct virulence profiles and genotypes [94,95]. In fact, it has been demonstrated that at least seven genotypes of *S. brasiliensis* are currently circulating in the Brazilian territory [95,96] and that these different genotypes can vary in terms of virulence profiles and susceptibility to antifungal drugs. Considering the incidence of different clinical patterns in distinct Brazilian states, which can be associated with the genotypes and the evolution of fungi from different environmental conditions, in addition to the well-known importance of cats on the sporotrichosis epidemiological chain, a One Health approach is necessary together with the investigation of clinical patterns in humans and distinct genotypes of *S. brasiliensis* to clarify these points.

A potential limitation of our systematic review was the bias caused by descriptor delimitation and by the non-registration of the systematic review before the beginning of the methods. In addition, the impossibility of recovering other nonscientific material using databases, like epidemiological bulletins or informal notices, which could report more cases from the same or other areas, is highlighted as an additional limitation.

HRs, with a predominance of erythema (nodosum or multiforme), described in patients from four Brazilian states (Rio de Janeiro, São Paulo, Rio Grande do Sul, and Pernambuco), have been associated with continuous exposure to the *S. brasiliensis* antigen, resulting in immune sensitization by the patients. The exposure to high levels of *S. brasiliensis* antigens commonly occurs in hyperendemic areas of zoonotic sporotrichosis. Close and prolonged contact of patients with lesions that are rich in fungal propagules from infected cats contribute to the overexpression of an immunological reaction during the disease development, which probably explains the high number of HRs in patients with zoonotic sporotrichosis in Brazil [12,23]. However, the exact immunological mechanism that promotes this inflammatory exacerbation still needs to be unveiled, contributing to research in the immune field of sporotrichosis [97]. In particular, the role of the host defense in determining disease predisposition is currently unknown, along with the protagonism of variables such as genetic predisposition, inoculum size, genotype, and *S. brasiliensis* virulence factors. Although HRs have a good prognosis and are not potentially severe, they need to be promptly and correctly diagnosed for the adequate management and treatment of the cases. 

Mucosal sporotrichosis, a prevalent condition that usually affects immunocompetent patients, occurs mainly as conjunctivitis, dacryocystitis, and Parinaud syndrome. These unspecific manifestations argue for the necessity of ophthalmologists’ awareness to include the infection by *S. brasiliensis* in the roll of the diagnostic hypotheses in patients from endemic areas. The clinical features are also distinct from ocular sporotrichosis caused by *S. schenckii* and *S. globosa*, which often manifest as eyelid lesions after an accidental traumatic fungal inoculation [98,99]. An explanation for this difference can be attributed to the source of infection, since in zoonotic sporotrichosis the mucosal surface is directly infected through contact with fungal propagules carried by sneezes of infected cats, which can carry a high quantity of fungal cells of *S. brasiliensis* [20,30,66,100]. 

As demonstrated in this systematic review, osteoarthritis caused by *S. brasiliensis* was an important atypical manifestation of zoonotic sporotrichosis, resulting in relevant impacts on the quality of life and prognosis of the patients [22,37,50,56,76,79,81,83]. Its consequences, like partial or total amputation, are unfortunately also common in patients with other neglected fungal diseases, such as mycetoma and chromoblastomycosis, and these are often associated with a late diagnosis [101]. Bone sporotrichosis was rarely reported until the emergence of the species *S. brasiliensis* and is frequently associated with various degrees of immunosuppression. Considering that 59 cases of osteoarthritis were published in Brazil over 24 years (1998 to 2021), while a previous review of the literature (1980–2015) found 20 cases over 36 years [33], we can hypothesize that *S. brasiliensis* is associated with more severe manifestations of sporotrichosis, in comparison to other species of *Sporothrix* [14,33]. This hypothesis is supported by in vivo studies. Arrillaga-Moncrieff et al. [102] showed a more expressive virulence of *S. brasiliensis* in a murine model in comparison with *S. globosa* and *S. schenckii*, stimulating a more severe form of sporotrichosis with the highest rate of dissemination and more severely cutaneous lesions. In addition, an in vivo study with different isolates recovered from one unique human patient in different periods showed that *S. brasiliensis* can increase its virulence during pathogenesis [103].

In the same line, systemic sporotrichosis with the involvement of internal organs (brain and lungs) has severe patterns of sporotrichosis and results in a poor prognosis, warning of a worrisome situation in Brazil [34,36,66]. Pulmonary sporotrichosis is, fortunately, still considered a rare condition, but it is associated with unspecific symptoms that overlap with other lung infections (including tuberculosis and histoplasmosis) [104]. This factor contributes to delays in its diagnosis, potentially leading to severe lung parenchyma destruction, which is associated with high rates of death, bringing the necessity to add sporotrichosis to the differential diagnosis of pulmonary fungal infections, particularly in hyperendemic areas in Brazil [72]. 

Meningitis caused by *S. brasiliensis* also showed unspecific symptoms (i.e., headache, mental confusion, weight loss, lethargy, and vomiting), similar to other meningeal infections, including tuberculosis, syphilis, and cryptococcosis [105]. Meningitis usually follows hematogenous dissemination, mostly in immunosuppressed patients. However, it may also occur in immunocompetent patients as a primary disease with brain involvement, which brings the necessity of in vivo studies to evaluate if some *S. brasiliensis* strains could have a primary neurological tropism, which has already been described in other fungi such as *Cryptococcus neoformans* [106,107] and evidenced in in vivo experimental studies [108,109]. 

The high rate of deaths and sequelae evidenced after the compilation and analyses of ~800 cases in our study can be attributed to poor host conditions, such as immunosuppression and other co-infections and comorbidities, and may have been aggravated by the unfamiliarity of health professionals with the unusual clinical presentations of sporotrichosis, with delayed diagnosis and treatment. Thus, it is urgently necessary to invest in educational programs for health professionals and the general population regarding zoonotic sporotrichosis, to increase the knowledge about individual control and preventive measures and to reduce underdiagnosis of patients. In addition, it is necessary to make efforts to continue educational activities to highlight the necessity of including sporotrichosis as a differential diagnosis of these atypical clinical manifestations in hyperendemic areas [110,111,112]. In addition, in the Brazilian territory, sporotrichosis is a disease of non-compulsory notification, which certainly makes cases underestimated; thus, our compiled cases are only a part of the real cases. It is necessary to implement public politics at a national level to know the real scenario of sporotrichosis in Brazil. 

Another gap to work with is the development of novel diagnostic tests for sporotrichosis, since cultures (current gold standard) require 7–21 days for a conclusive positive result [12,113]. Even though such a delay in non-severe cases may not be a problem, in systemic cases this can directly influence the prognosis, sequelae, and outcome of patients. Therefore, the development of rapid tests is urged to diagnose extracutaneous and atypical cases of zoonotic sporotrichosis, being a promising field for research and innovation.

## 5. Conclusions

In conclusion, here we have compiled and summarized data on atypical zoonotic sporotrichosis, showing that severe and unusual manifestations of this mycosis are not uncommon and are geographically dispersed in the Brazilian territory, supporting previous in vivo and clinical studies that hypothesize that *S. brasiliensis* is a species that is more virulent than sapronotic *Sporothrix* species. Therefore, educational measures are needed to make health professionals and the general population aware of different zoonotic sporotrichosis manifestations that can occur in endemic areas of Brazil. In addition, research should evolve to improve diagnosis, aiming for a quicker diagnostic method in severe cases, and the role of new antifungal agents should also be evaluated. A better understanding of the relevance of *S. brasiliensis* genotypes in relation to these diverse clinical presentations is ultimately needed.

## Figures and Tables

**Figure 1 jof-10-00287-f001:**
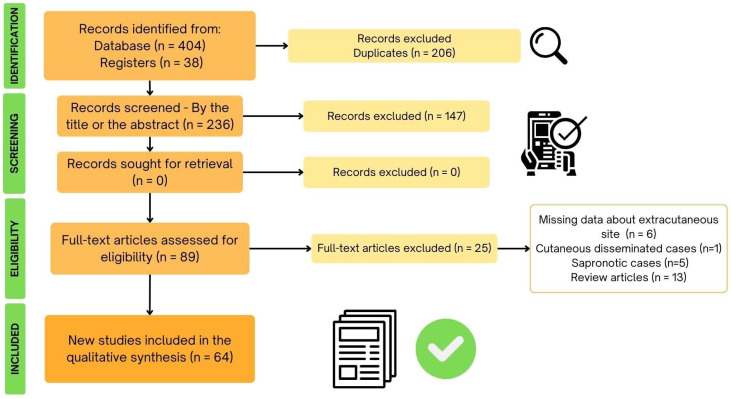
Flowchart describing the total of scientific articles obtained by the database searches for atypical cases of sporotrichosis by *Sporothrix brasiliensis* that occurred in Brazil and included in this study.

**Figure 2 jof-10-00287-f002:**
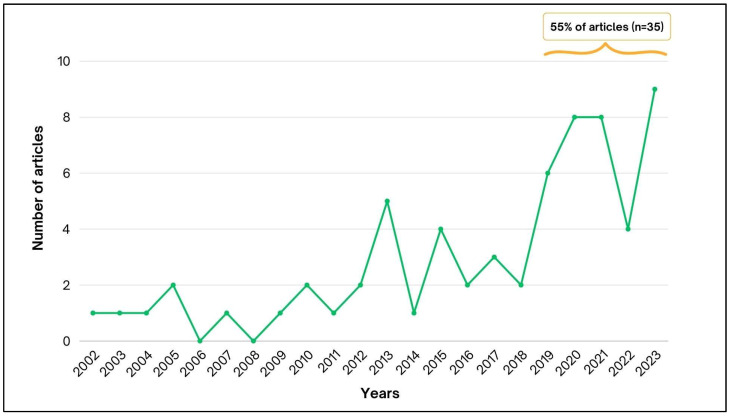
Temporal distribution of articles describing atypical zoonotic sporotrichosis caused by *Sporothrix brasiliensis* between the years of 2002 and 2023, highlighting an increase of 700% in the number of publications comparing two periods of five years (beginning and most recent period).

**Figure 3 jof-10-00287-f003:**
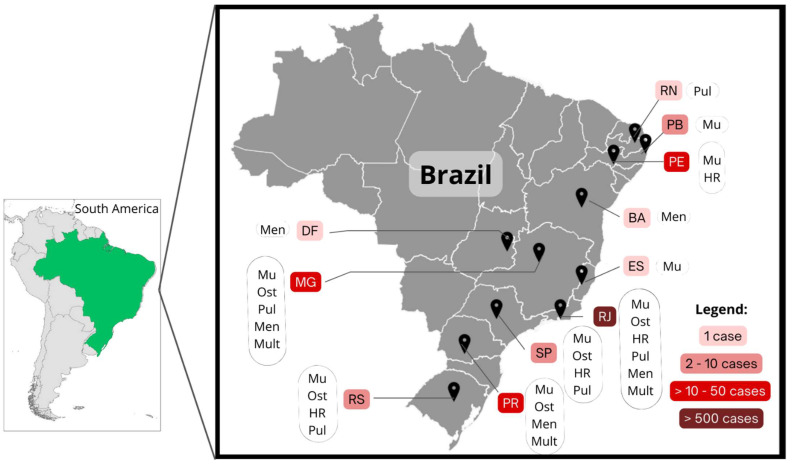
South America map highlighting (green color) the only country (Brazil) where atypical zoonotic sporotrichosis has been reported and a Brazilian map showing the distribution of cases reported through states. DF: Distrito Federal; MG: Minas Gerais; RS: Rio Grande do Sul; PR: Paraná; SP: São Paulo; RJ: Rio de Janeiro; ES: Espírito Santo; BA: Bahia; PE: Pernambuco; PB: Paraíba; RN: Rio Grande do Norte. Mu: mucosal; Ost: osteoarthritis; HR: hypersensitivity reaction; Pul: pulmonary; Men: meningeal; Mult: multifocal.

**Table 1 jof-10-00287-t001:** Extracutaneous zoonotic sporotrichosis data.

Extracutaneous Type	Clinical Manifestation/Site of Infection	Number	Immunosuppression/ Comorbidities (%)	Primary Site of Infection (%)	Outcome (%)	Refs.
Hypersensitivity		370	Yes 11/29 (38%) No 18/29 (62%)	Not applicable	Cure 318/318 (100%)	[14,17,24,25,26,27,28,42,47,54,69,76,81]
	Erythema nodosum	184				
	Erythema multiforme	143				
	Sweet syndrome	35				
	Arthritis	8				
Mucosal		256	Yes 29/65 (45%) No 36/65 (55%)	Yes 229/250 (92%) No 21/250 (8%)	Cure 193/193 (100%); ^#^ Sequelae 23/193 (12%)	[15,20,21,29,31,32,41,42,47,48,52,53,57,60,61,63,64,65,67,70,71,73,74,75,78,79,81,82]
	Ocular	221				
	Nasal	28				
	Oral	2				
	Ocular and nasal	1				
	Oral and nasal	4				
Osteoarthritis	Hand Upper and lower limb Foot and/or ankle Knee Elbow Clavicle	59 26 9 6 3 1 1	Yes 26/28 (93%) No 2/28 (7%)	Yes 22/49 (45%) No 27/49 (55%)	Cure 22/26 (85%); ^##^ Sequelae 3/22 (14%) Death 4/26 (15%)	[22,37,46,50,56,76,79,80,81,83]
Meningeal		32	Yes 29/32 (91%) No 3/32 (9%)	Yes 3/32 (9%) No 29/32 (91%)	Cure 9/30 (30%) ^###^ Sequelae 1/9 (11%); Death 21/30 (70%)	[19,34,35,36,43,44,45,47,55,62]
Pulmonary	Cavitary nodules Bronchiectasis Infiltrate Pleural effusion Fibrosis	14 4 2 1 1 1	Yes 7/9 (78%) No 2/9 (22%)	Yes 10/14 (71%) No 4/14 (29%)	Cure 3/5 (60%) Death 2/5 (40%)	[18,37,49,51,59,66,68,72,77,81]
Multifocal		60	Yes 46/46 (100%) No 0/46 (0%)	Not applicable	Cure 19/35 (54%) ^####^ Sequelae 2/19 (11%); Death 16/35 (46%)	[18,19,21,22,37,47,58,71,79]
	* Two sites	53				
	** Three sites	6				
	*** Four sites	1				

^#^ Mucosal sequelae: ocular (chronic dacryocystitis, corneal changes, cutaneous fistula, lagophthalmos, ectropion, entropion, pannus 180°, symblepharon, conjunctival fibrosis, paracentral leucoma, or eyelid retraction) or nasal (hyperrhynolalia and retraction of the right ala nasi); ^##^ sequelae osteoarthritis: amputation, total or partial; ^###^ meningeal sequelae: neurological sequelae (ataxia and extrinsic ocular motor paresis related to basal meningitis); ^####^ multifocal sequelae: amputation. * Two sites: osteoarthritis + mucosal (n = 30); osteoarthritis + pulmonary (n = 13); hypersensitivity + meningeal (n = 6); osteoarthritis + meningeal (n = 3); meningeal + pulmonary (n = 2); mucosal + meningeal (n = 2); hypersensitivity + mucosal (n = 1). ** Three sites: osteoarthritis + meningeal + pulmonary (n = 3); osteoarthritis + meningeal + mucosal (n = 2); osteoarthritis + mucosal + pulmonary (n = 1). *** Four sites: osteoarthritis + mucosal + meningeal + pulmonary. REF: reference.

**Table 2 jof-10-00287-t002:** Outcome of multifocal cases of extracutaneous zoonotic sporotrichosis.

Number of Sites	Types	N	Outcome (%)	Refs.
Two	Osteoarthritis + mucosal	32	Cure (68%; 13/19) Death (32%; 6/19)	[18,19,21,22,37,47,58,71,79]
	Osteoarthritis + pulmonary	14	Cure (67%; 6/9) Death (33%; 3/9)	
	Osteoarthritis + meningeal	3	Death (100%; 2/2)	
	Meningeal + pulmonary	2	Death (100%; 1/1)	
	Mucosal + meningeal	2	---	
Three	Osteoarthritis + meningeal + pulmonary	3	Death (100%; 2/2)	
	Osteoarthritis + meningeal + mucosal	2	Death (100%; 1/1)	
	Osteoarthritis + mucosal + pulmonary	1	Death (100%; 1/1)	
Four	Osteoarthritis + mucosal + meningeal + pulmonary	1	---	

N: number; Refs.: reference.

## Data Availability

Data sharing is not applicable to this article.

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
