# Peer review of "Sporothrix brasiliensis Causing Atypical Sporotrichosis in Brazil: A Systematic Review"

_jof, 2024, doi:10.3390/jof10040287_

Round 1

Reviewer 1 Report

This is a very interesting review about atypical cases of zoonotic sporotrichosis due to Sporothrix brasiliensis. The distribution of cases is well described, and clinical manifestations showed a high percentage of hypersensitivity reactions.

As all the cases only correspond to Brazilian patients, I consider the title could be changed to:  Sporothrix brasiliensis causing atypical sprotrichosis in Brazil: A systematic review.

If the data is approximated to whole numbers instead of 4.5% it should read 5% (line 153). The same happens in lines 186 and 187 (mucosal cases).

Author Response

Revision Note

April 2024

Dear Prof. Dr. David S. Perlin

Editor-in-chief of Journal of Fungi

Thank you for considering our manuscript entitled “Sporothrix brasiliensis causing atypical sporotrichosis in Brazil: A systematic review for publication. We have added all the modifications (described below and highlighted in red in the text) required by the reviewers.

We expect this revised version of our manuscript is now suitable for publication in Journal of Fungi.

Sincerely yours,

Vanice Rodrigues Poester, Melissa Orzechowski Xavier, Lívia Silveira Munhoz, Rossana Patricia Basso, Rosely Maria Zancopé-Oliveira, Dayvison Francis Saraiva Freitas, Alessandro Comarú Pasqualotto

Reviewer #1:

Major comments: This is a very interesting review about atypical cases of zoonotic sporotrichosis due to Sporothrix brasiliensis. The distribution of cases is well described, and clinical manifestations showed a high percentage of hypersensitivity reactions.

Thank you for the evaluation and comments.

Detail comments: As all the cases only correspond to Brazilian patients, I consider the title could be changed to:  Sporothrix brasiliensis causing atypical sprotrichosis in Brazil: A systematic review.

          In agreement with the Reviewer comment, the title was changed.

If the data is approximated to whole numbers instead of 4.5% it should read 5% (line 153). The same happens in lines 186 and 187 (mucosal cases).

Corrected.

Reviewer 2 Report

Overall Statement

Despite the commendable quality of the work, a significant concern arises from the methodological section. It is with this in mind that we propose a major revision. In other aspects, the study is exemplary.

Materials and Methods

Page 3, Line 98: "Please revise the entirety of this section in accordance with the most recent PRISMA guidelines [1]. Additionally, furnish the registration number for this systematic review. In the absence of such registration, a significant source of bias is introduced."

Results

Page 3, Line 129: "This section requires revision to align with the latest PRISMA guidelines [1], including adjustments to Figure 1."

Discussion

Public Health Implications: "While the discussion addresses the necessity for educational initiatives and enhanced diagnostic procedures, it would benefit from an expanded exploration of specific public health strategies or interventions aimed at curbing the spread of atypical sporotrichosis. Potential recommendations could encompass surveillance, awareness campaigns, or education for healthcare providers."

Limitations

"It is recommended to introduce a distinct section dedicated to limitations, enabling the identification and comprehensive discussion of such constraints."

References

Page MJ, McKenzie JE, Bossuyt PM, Boutron I, Hoffmann TC, Mulrow CD, Shamseer L, Tetzlaff JM, Akl EA, Brennan SE, Chou R, Glanville J, Grimshaw JM, Hróbjartsson A, Lalu MM, Li T, Loder EW, Mayo-Wilson E, McDonald S, McGuinness LA, Stewart LA, Thomas J, Tricco AC, Welch VA, Whiting P, Moher D. The PRISMA 2020 statement: an updated guideline for reporting systematic reviews. BMJ. 2021 Mar 29;372:n71. doi: 10.1136/bmj.n71. PMID: 33782057; PMCID: PMC8005924.

Author Response

Revision Note

April 2024

Dear Prof. Dr. David S. Perlin

Editor-in-chief of Journal of Fungi

Thank you for considering our manuscript entitled “Sporothrix brasiliensis causing atypical sporotrichosis in Brazil: A systematic review for publication. We have added all the modifications (described below and highlighted in red in the text) required by the reviewers.

We expect this revised version of our manuscript is now suitable for publication in Journal of Fungi.

Sincerely yours,

Vanice Rodrigues Poester, Melissa Orzechowski Xavier, Lívia Silveira Munhoz, Rossana Patricia Basso, Rosely Maria Zancopé-Oliveira, Dayvison Francis Saraiva Freitas, Alessandro Comarú Pasqualotto

Reviewer #2:

Major comments: Despite the commendable quality of the work, a significant concern arises from the methodological section. It is with this in mind that we propose a major revision. In other aspects, the study is exemplary.

          Thank you for your comments. In the following, we seek to clarify the important points considered.

Materials and Methods: Page 3, Line 98: "Please revise the entirety of this section in accordance with the most recent PRISMA guidelines [1]. Additionally, furnish the registration number for this systematic review. In the absence of such registration, a significant source of bias is introduced."

          Methods were revised and we added a phrase showing the lack of the registration number as a limitation of our study.

Results: Page 3, Line 129: "This section requires revision to align with the latest PRISMA guidelines [1], including adjustments to Figure 1."

          Figure 1 was corrected, and the section was revised.

 Discussion: Public Health Implications: "While the discussion addresses the necessity for educational initiatives and enhanced diagnostic procedures, it would benefit from an expanded exploration of specific public health strategies or interventions aimed at curbing the spread of atypical sporotrichosis. Potential recommendations could encompass surveillance, awareness campaigns, or education for healthcare providers."

          In agreement with the suggestion, a phrase in this field was included in the Discussion section.

Limitations: "It is recommended to introduce a distinct section dedicated to limitations, enabling the identification and comprehensive discussion of such constraints."

In agreement with the suggestion, limitations were included in the Discussion section.

Reviewer 3 Report

Dear authors,

Your review article reports extracutaneous presentations associated with zoonotic sporotrichosis and hypersensitivity reactions. The paper, as a review, summarises the current data on the subject with clear tables and figures. The topic is exciting and actual from a Global Perspective (Global Health) and the One Health point of view. Major points are the high self-plagiarism and the research strategy (including not-proven infections). I report some notes to improve the review.

1)      Despite this paper being a review, with specific inclusion and exclusion criteria in the research strategy, the percentage of self-citation is too high. I suggest that the author look at the works of Hector Mora Montes, Maria Clara Gutierrez Galhardo and Rodrigo Almeida-Paes to reduce the self-citation percentage or reduce their own references if not directly pertinent to the research strategy.

2)      In the abstract, you start in medias res. It is opportune to report a few lines about sporotrichosis, definition, geographical context and why your paper is innovative.

3) The Prisma checklist should be uploaded as non-published supplementary files and revised.

4)      Explain the difference between atypical and extracutaneous sporotrichiosis. You should unify the terminology in the main file.

5)      In the tables, you should report if the article reports a proven or a possible sporotrichosis. Moreover, it would be better to remove the possible case (that can be considered a bias) to have a higher level of evidence. I suggest that the authors to evaluate all the articles found and separate the proven diagnosis and the possible diagnosis. Moreover, report guidelines to the correct definition of probable and possible.

6)      Why did you select 1990 as a starter point?

7)      Excel version is missing

8)      In the figure 2, the orange arrow can be removed.

9)      Results section should include subparagraphs based on the examined points.

10)   Lines 166-250 report clinical and demographic data not presented in the tables. You should report a table with the demographic data or improve tables 1 and 2 with the patients’ features (age, sex, etc). Moreover, the standard deviation in patients’ demographic data is missing.

11)   In table 2 there are missing references.

12)   I suggest reporting, in the discussion section, a few lines about One Health and Global Health and why sporotrichosis is a hot point of these themes, as well as the fast microbiology.

13)   Study limitations are missing.

14)   Line 346, in vivo italicized please.

Author Response

Revision Note

April 2024

Dear Prof. Dr. David S. Perlin

Editor-in-chief of Journal of Fungi

Thank you for considering our manuscript entitled “Sporothrix brasiliensis causing atypical sporotrichosis in Brazil: A systematic review for publication. We have added all the modifications (described below and highlighted in red in the text) required by the reviewers.

We expect this revised version of our manuscript is now suitable for publication in Journal of Fungi.

Sincerely yours,

Vanice Rodrigues Poester, Melissa Orzechowski Xavier, Lívia Silveira Munhoz, Rossana Patricia Basso, Rosely Maria Zancopé-Oliveira, Dayvison Francis Saraiva Freitas, Alessandro Comarú Pasqualotto

Reviewer #3:

Major comments: Dear authors, Your review article reports extracutaneous presentations associated with zoonotic sporotrichosis and hypersensitivity reactions. The paper, as a review, summarises the current data on the subject with clear tables and figures. The topic is exciting and actual from a Global Perspective (Global Health) and the One Health point of view. Major points are the high self-plagiarism and the research strategy (including not-proven infections). I report some notes to improve the review.

          Thank you for your comments. In the following, we seek to clarify the important points considered.

Detail comments: 1) Despite this paper being a review, with specific inclusion and exclusion criteria in the research strategy, the percentage of self-citation is too high. I suggest that the author look at the works of Hector Mora Montes, Maria Clara Gutierrez Galhardo and Rodrigo Almeida-Paes to reduce the self-citation percentage or reduce their own references if not directly pertinent to the research strategy.

          The systematic review was designed by the Rio Grande do Sul group which has projects in partnership with researchers from Rio de Janeiro. Rio de Janeiro state is the epicenter of zoonotic sporotrichosis in Brazil, and the co-author researchers (Dr. Rosely Maria Zancopé-Oliveira and Dr. Dayvison Francis Saraiva Freitas) are some references in studies describing clinical and epidemiological data of this field. Thus, we can´t avoid self-citation, since that the majority of articles described in the literature were published by the groups that are in the authorship of this paper. We highlighted that Maria Clara Gutierrez Galhardo and Rodrigo Almeida-Paes were already contemplated in the References lits, and we added citation of important studies form Dr. Hector Mora Montes's, in completely agreement with the Reviewer's comment.

2)      In the abstract, you start in medias res. It is opportune to report a few lines about sporotrichosis, definition, geographical context and why your paper is innovative.

          The introduction of the abstract was improved.

3) The Prisma checklist should be uploaded as non-published supplementary files and revised.

          Included.

4)      Explain the difference between atypical and extracutaneous sporotrichiosis. You should unify the terminology in the main file.

          Atypical sporotrichosis includes all forms that are not cutaneous forms (fixed and lymphocutaneous), being mucosal, hypersensitivity reactions (HR), osteoarthritis, pulmonary, and meningeal. The extracutaneous definition includes all of these except HR, which is an immune response and not an extracutaneous manifestation of sporotrichosis. An explanation was included in the methods. However, we do not unify the terminology, considering the difference between the words. 

5)      In the tables, you should report if the article reports a proven or a possible sporotrichosis. Moreover, it would be better to remove the possible case (that can be considered a bias) to have a higher level of evidence. I suggest that the authors to evaluate all the articles found and separate the proven diagnosis and the possible diagnosis. Moreover, report guidelines to the correct definition of probable and possible.

Criteria to classify zoonotic sporotrichosis as probable and proven has been recently published (Queiroz-Telles et al., 2022). Since it is robust and has a good acceptance in the international literature (Ministério da Saúde, Brazil, 2023; Cognialli et al., 2023), we decided to maintain the probable cases in our revision. We added the citation in the Methods. In addition, considering that most cases were proven, as reported in the Results ("…and in 98% a proven diagnosis of sporotrichosis was achieved by the isolation of the fungus in mycological culture"), and in Tables, the data was better shown as a compiled of different manifestations, being, in our opinion, the discrimination of proven and probable sporotrichosis unnecessary.

Reference:

Queiroz-Telles F, Bonifaz A, Cognialli R, Lustosa BPR, Vicente VA, Ramírez-Marín HA. Sporotrichosis in Children: Case series and Narrative Review. Curr Fungal Infect Rep. 2022;16(2):33-46. doi:10.1007/s12281-022-00429-x

Cognialli, R. C. R.; Cáceres, D. H.; Bastos, F. de A. G. D.; Cavassin, F. B.; Lustosa, B. P. R.; Vicente, V. A.; Breda, G. L.; Santos-Weiss, I.; Queiroz-Telles, F. Rising Incidence of Sporothrix Brasiliensis Infections, Curitiba, Brazil, 2011–2022. Emerging Infectious Diseases 2023, 29 (7). https://doi.org/10.3201/eid2907.230155.

Ministério da Saúde, Brazil. Nota técnica acerca de recomendações sobre a vigilância da esporotricose animal no Brasil. Ofício Circular nº 102/2023/SVSA/MS. 2023.

6)      Why did you select 1990 as a starter point?

This decision was based in the fact that the 90’s was the decade when the first outbreaks of zoonotic sporotrichosis started to be described and to grow exponentially, turning the sapronotic cases secondary in importance and frequency. We included this information in the Methods section.

7)      Excel version is missing

          The information was included. 

8)      In the figure 2, the orange arrow can be removed.

          In agreement with the suggestion, Figure 2 was improved.

9)      Results section should include subparagraphs based on the examined points.

In agreement with the suggestion, subtopics were included.

10)   Lines 166-250 report clinical and demographic data not presented in the tables. You should report a table with the demographic data or improve tables 1 and 2 with the patients’ features (age, sex, etc). Moreover, the standard deviation in patients’ demographic data is missing.

We decided to include in the Tables only the most relevant data to classify the type of sporotrichosis (clinically) and its impact (outcome). Standard deviation was included.

11)   In table 2 there are missing references.

References in Table 2 were together to all data, we checked and did not find the missing reference. Can you kindly indicate the missing reference?

12)   I suggest reporting, in the discussion section, a few lines about One Health and Global Health and why sporotrichosis is a hot point of these themes, as well as the fast microbiology.

In agreement with the suggestion, we included a phrase regarding this theme in the Discussion section.

13)   Study limitations are missing.

          In agreement with the suggestion, limitations were included in the Discussion section.

 14)   Line 346, in vivo italicized please.

          Corrected.

Reviewer 4 Report

The legend of Fig. 1 Specify that it was only from the country Brazil.

Is the fact that the majority of cases are in the last 5 years because cases have increased or because they have been reported and were not reported before?  Fig. 2

Is it known or is there any hypothesis as to why the majority of cases occur in RJ. Is it due to tourism or the greater number of felines or due to neglect of stray cats?

I consider this point very important. Expand a little more. Lines: 329-331

I suggest an additional figure, where the most affected places in the human body by said mycosis are quickly appreciated.

Resultado de traducción

Author Response

Revision Note

April 2024

Dear Prof. Dr. David S. Perlin

Editor-in-chief of Journal of Fungi

Thank you for considering our manuscript entitled “Sporothrix brasiliensis causing atypical sporotrichosis in Brazil: A systematic review for publication. We have added all the modifications (described below and highlighted in red in the text) required by the reviewers.

We expect this revised version of our manuscript is now suitable for publication in Journal of Fungi.

Sincerely yours,

Vanice Rodrigues Poester, Melissa Orzechowski Xavier, Lívia Silveira Munhoz, Rossana Patricia Basso, Rosely Maria Zancopé-Oliveira, Dayvison Francis Saraiva Freitas, Alessandro Comarú Pasqualotto

Reviewer #4:

MAJOR: The legend of Fig. 1 Specify that it was only from the country Brazil.

The information was added to the legend of Figure 1.

 Is the fact that the majority of cases are in the last 5 years because cases have increased or because they have been reported and were not reported before?  Fig. 2

          Figure 2 refers to the increase in articles published, but also an expressive increase in zoonotic sporotrichosis cases occurred, this information is exposed in the introduction section.

 Is it known or is there any hypothesis as to why the majority of cases occur in RJ. Is it due to tourism or the greater number of felines or due to neglect of stray cats?

Zoonotic sporotrichosis emerged in the 90s in the states of Rio de Janeiro and Rio Grande do Sul. However, in RJ a quickly epidemic established, and nowadays, the state is the epicenter of zoonotic sporotrichosis with thousands of cases described each year (Giordano, 2021). Maybe, this is one of the reasons for it to being the origin of the majority of atypical cases. Another hypothesis is related to the most prevalent genotype of the S. brasiliensis circulating in the state which could be more virulent than others (Rabello et al., 2024). A greater number of stray cats is the reality of several Brazilian states and indeed this condition did auxiliary emergence of sporotrichosis as a severe public health issue in Brazil.

Reference:

Giordano C. BOLETIM EPIDEMIOLÓGICO ESPOROTRICOSE Nº 001/2021. Estado do Rio de Janeiro. 2021

Rabello VBS, de Melo Teixeira M, Meyer W, Irinyi L, Xavier MO, Poester VR, Pereira Brunelli JG, Almeida-Silva F, Bernardes-Engemann AR, Ferreira Gremião ID, Dos Santos Angelo DF, Clementino IJ, Almeida-Paes R, Zancopé-Oliveira RM. Multi-locus sequencing typing reveals geographically related intraspecies variability of Sporothrix brasiliensis. Fungal Genet Biol. 2024 Feb;170:103845. doi: 10.1016/j.fgb.2023.103845.

I consider this point very important. Expand a little more. Lines: 329-331

In agreement with the suggestion, we improved the theme discussed.

I suggest an additional figure, where the most affected places in the human body by said mycosis are quickly appreciated.

The information requested by the Reviewer was already available in the text and Table 1.

Round 2

Reviewer 2 Report

None

All comments have been adequately addressed, with only a minor point remaining. A quality assessment of each article is still needed to complete the review.

Reviewer 3 Report

Dear Authors,

all the points have been addressed and the manuscript has been improved.

Dear Authors,

all the points have been addressed and the manuscript has been improved.